green chemistry/organic chemistry/materials science

waste vegetable oil, alkylation, strong acid cation exchange resins, aryl fatty acids derivatives, deactivation mechanism, Diels–Alder cycloaddition

**Author for correspondence:**
Shi-Zhong Yang
e-mail: meor@ecust.edu.cn

This article has been edited by the Royal Society of Chemistry, including the commissioning, peer review process and editorial aspects up to the point of acceptance.

# The optimization of heterogeneous catalytic conditions in the direct alkylation of waste vegetable oil

Cheng-Long Gao, Xin Wang, Hong-Ze Gang, Jin-Feng Liu, Bo-Zhong Mu and Shi-Zhong Yang

State Key Laboratory of Bioreactor Engineering and Institute of Applied Chemistry, East China University of Science and Technology, Shanghai 200237, People's Republic of China

C-LG, 0000-0003-2429-5926; S-ZY, 0000-0001-9979-4469

Alkylated waste vegetable oil is a versatile intermediate product in the synthesis of bio-based materials. Heterogeneous catalytic condition with high conversion rate in the direct alkylation of waste vegetable oil was reported and the deactivation mechanism of catalyst was revealed. The total exchange capacity, elemental composition and pyrolysis product of catalyst before and after the alkylation reaction were analysed by back titration, elemental analysis, electrospray ionization mass spectrometry, gas chromatography mass spectrometry and pyrolysis-gas chromatography/mass spectrometry, respectively. The results indicated that the metallic and non-metallic (C, H) elements contents of the catalyst have very much increased with great changes in pyrolysis product and a slight decrease in the total exchange capacity. The formation of insoluble polymers through Diels–Alder cycloaddition between triglycerides was proved to be the major factor causing the dysfunction of the catalytic centre. The metal ions from corrosion of the reactor were the minor factor causing about 2.56% loss of the catalytic centre. Moreover, the catalyst was able to maintain high catalytic efficiency when replacing the raw materials with other waste vegetable oil having low concentration of polyunsaturated fatty acids, which is significant for producing not only the aryl fatty acids derivatives but also the bio-based surfactants.

## 1. Introduction

Waste vegetable oil is a series of oils banned in food industry, including waste cooking oil, non-edible oil and expired oil. Utilization of waste vegetable oil for value-added products has become a hot research topic in recent years due to the illegal usage of the gutter oil in food industry [1–4]. Through simple

**Figure 1.** The comparison of two processes and this work in alkylation of waste vegetable oil.

reactions, the waste vegetable oil can be converted into other versatile substances, such as biodiesel [5–7], fatty acids derivatives [8–10] and epoxidized soya bean oils [11–14]. In addition, bio-based functional materials can be produced through the chemical modification of waste vegetable oil, such as functional copolymers [15–17] and surfactants [18–20].

In the synthesis of surfactants, alkylated oil is an important intermediate which can introduce special function to the surfactants. The surface/interfacial properties are much better when using the alkylated oil in the synthesis of zwitterionic surfactants [21–25]. Traditional route to synthesize the alkylated oil involves the pretreatment of waste vegetable oil, such as hydrolysis [24,26] and trans-esterification [27] (figure 1). In addition, catalyst used in the traditional synthesis route is homogeneous catalyst which is hard to be separated and recycled. The research into the direct alkylation of waste vegetable oil was still limited [28,29].

Herein, this paper presented a heterogeneous catalytic condition in the direct alkylation of waste vegetable oil and the deactivation mechanism was explored in detail.

Strong acid cation exchange resins were purchased from Nanda Synthesis Co., Ltd (Jiangsu, China) and the Aladdin Chemical Reagent (Shanghai, China). Waste vegetable oil collected from canteens and restaurants was firstly pretreated through filtration and centrifugation. Then the upper oil phase was acidified, filtered and washed with water. Finally, the starting material was obtained after dehydration.

Five strong acid cation exchange resins were tested as catalyst in the direct alkylation of waste vegetable oil and HND 580 exhibited highest catalytic efficiency (table 1). When the total exchange capacity was close to some extent, the relationship between pore capacity and conversion rate was positive correlation. When pore capacity was large, the catalyst was able to contain much more substrates and the exchange of substances with reaction solutions was easy. Consequently, pore capacity should be considered in choosing the ideal catalyst and HND 580 was the suitable catalyst in the direct alkylation of waste vegetable oil.

Using HND 580 as catalyst, the following optimized alkylation conditions were achieved, temperature $= 140°C$, solvent ratio $(v/v) = 3$, catalyst loading $(w/v) = 0.55$, reaction time $= 2$ h. F-test (variance ratio) and orthogonal analysis (electronic supplementary material, tables S1 and S2) showed that the temperature, catalyst loading and the reaction time were the main factors with the solvent ratio being the minor factor.

The catalyst in the alkylation was recycled for 20 rounds in the optimized condition to investigate the operational life of the catalyst. The catalytic efficiency decreased rapidly from 97.8% to 53.9% with the recycle times increasing (figure 2). In addition, the linear fitting curve of the experiment data was $y = 1.014 - 0.012x$ and the calculated half-life was 42.2 h, which is much shorter than the half-life of the traditional strong acid cation exchange resins. Thus, the catalyst must be deactivated and experiments should be performed to explore the deactivation mechanism in the direct alkylation of the waste vegetable oil.

The deactivation mechanism of the catalyst was attributed to the loss and dysfunction of catalytic centre. Previous studies [30–32] showed that the C–S bond and C–C bond in strong acid cation

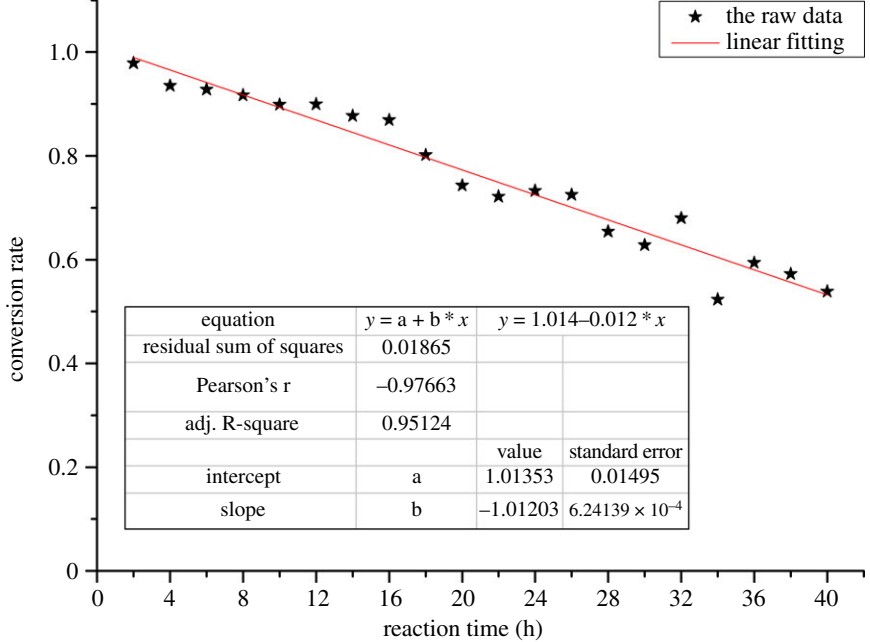

**Figure 2.** The effect of reaction time on the catalytic efficiency.

**Table 1.** The direct alkylation of waste cooking oil catalysed by strong acid cation exchange resins ($n = 3$).

| catalyst | pore capacity (ml/g) | capacity (mmol $g^{-1}$[$H^+$]) | max. op. temp (°C) | conversion rate (%) | yield (%) | selectivity (%) |
|---|---|---|---|---|---|---|
| Amberlyst 35 | 0.33 | 5.2 | 150 | 83.49 ± 1.57 | 28.06 ± 4.55 | 33.55 ± 4.86 |
| Amberlyst 36 | 0.2 | 5.4 | 150 | 67.96 ± 1.25 | 8.07 ± 1.16 | 11.88 ± 1.79 |
| HND 580 | 0.4 | 4.95 | 140 | 95.78 ± 2.43 | 76.80 ± 4.39 | 80.13 ± 2.62 |
| HND 2 | 0.3 | 4.75 | 120 | 23.12 ± 1.28 | 0.20 ± 0.09 | 0.87 ± 0.33 |
| Amberlyst 15 | 0.4 | 4.0 | 120 | 28.93 ± 0.60 | 0.56 ± 0.09 | 1.92 ± 0.27 |

exchange resins were thermally stable at 200°C. Thus, the loss of catalytic centre was mainly focused on the metal ions from corrosion of the reactor. Inductively coupled plasma atomic emission spectrometer (ICP) results (electronic supplementary material, table S3) showed that the metal ions in the recycled catalyst has increased largely when compared to metal ions in the unused catalyst. However, the contribution of the metal ions to the loss of total exchange capacity was only 2.56% after calculation. Thus, the metal ions from corrosion of the reactor was the minor factor and there must be another factor causing the declination of catalytic efficiency.

The total exchange capacity of the recycled catalyst was 3.82 mmol $g^{-1}$ [$H^+$], which was about 76.37% of the unused catalyst exchange capacity (electronic supplementary material, table S4). In addition, the content of sulfur had decreased significantly from 16.7% to 11.9% (electronic supplementary material, table S5). Thus, some extra substances were attached to the catalyst causing the declination of exchange capacity and the drop of the sulfur content. Furthermore, pyrolysis products of the catalyst were investigated, and its spectra can be divided into five parts according to the categories of the peaks (electronic supplementary material, S1). The peaks of sulfur dioxide, benzene, toluene, ethylbenzene and styrene were in the part 1 (A1, B1, C1), while the peaks of the fatty acids were in the part 5 (A5, B5, C5). As for the spectra of the recycled catalyst, the contents of the sulfur dioxide had decreased rapidly with an increase of aromatic hydrocarbons (containing sulfur, C2) and polycyclic aromatic hydrocarbons (C3). Besides, peaks in C5 indicated that little waste vegetable oil was left in the recycled catalyst and it was not the factor causing the change of spectra. As a consequence, some extra substances attached to the catalyst were the major factor causing the dysfunction of the catalytic centre, which was also confirmed by FTIR results (electronic supplementary material, figures S5 and S6).

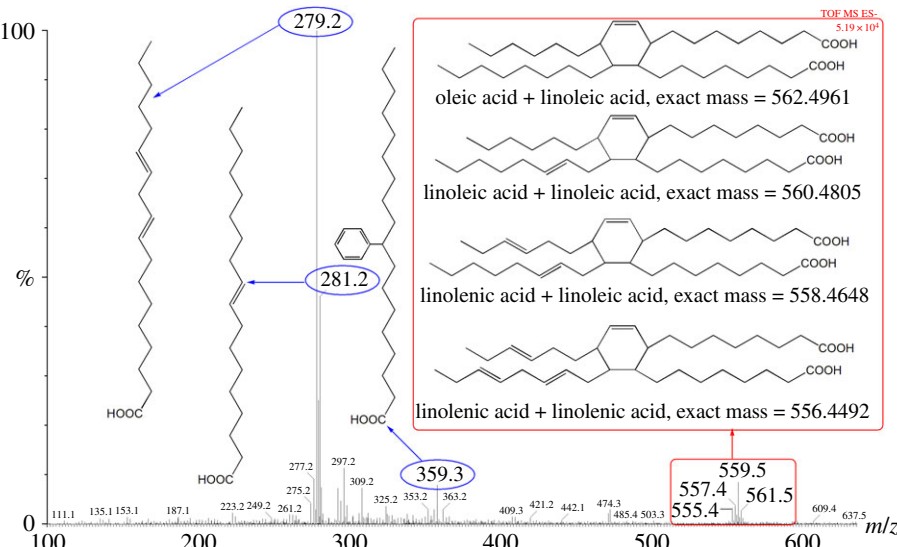

**Figure 3.** The ESI spectra of the extraction solution.

Herein, extraction by benzene and saponification by alkali were used to remove the extra substances attached to the catalyst. However, little substances were washed off the catalyst, indicating that the attachments were hard to dissolve in benzene and showed strong alkali resistance (electronic supplementary material, figures S2 and S3). As for the substances washed off the catalyst, in the electrospray ionization mass spectrometry (ESI-MS) spectra, there were a series of peaks from 553.4 to 567.4 (figure 3), which were properly the dimers of the unsaturated fatty acids. Furthermore, through Diels–Alder cycloaddition reaction between triglycerides, polymers were formed at high temperature [33] or catalysed by Lewis acid at lower temperature. Thus, the polymer formation through Diels–Alder cycloaddition reaction between triglycerides was the reason causing the dysfunction of the catalytic centre.

Diels–Alder cycloaddition reaction between triglycerides was based on the polyunsaturated fatty acids in the waste vegetable oil. In order to achieve high conversions and maintain excellent catalytic efficiency, waste vegetable oil was divided into three categories according to the contents of polyunsaturated fatty acids.

As for waste cooking oil, the contents of polyunsaturated fatty acids were very high, resulting in formation of insoluble polymers and declination of the catalytic efficiency as well as operational life. Two-step alkylation is necessary for this kind of waste vegetable oil to avoid the formation of insoluble polymers. Through hydrolysis or trans-esterification, waste vegetable oil can be converted to fatty acid derivatives which polymerize into soluble dimers and do not influence the operational life of the catalyst. For example, the half-life of the catalyst can be extended from 42.2 to 453 h (electronic supplementary material, figure S4) when using trans-esterification products of the waste vegetable oil in the preparation of alkylated waste vegetable oil.

As for olive oil, the content of oleic acid was high while the contents of polyunsaturated fatty acids were relatively low, resulting in mild formation of insoluble polymers and slow declination of the catalytic efficiency as well as operational life.

As for hydrogenated castor oil, 12-hydroxystearic acid could be used in the alkylation of hydrogenated castor oil, avoiding the formation of polymers. Thus, the hydrogenated castor oil will be an ideal raw material in the direct alkylation of waste vegetable oil.

## 2. Conclusion

Waste vegetable oil was efficiently converted to valuable alkylated oil using strong acid exchange resins as catalyst under optimized conditions. The insoluble polymers formation through Diels–Alder cycloaddition reaction between triglycerides was proved to be the major factor causing the dysfunction of the catalytic centre and the declination of the catalytic efficiency. Only about 2.56% loss of catalytic centre was caused by metal ions from the corrosion of the reactor. In addition, the

heterogeneous catalyst could maintain high catalytic efficiency when replacing the raw materials with other waste vegetable oil having low abundance of polyunsaturated fatty acids. Above all, this paper illustrated heterogeneous catalytic condition in the direct alkylation of waste vegetable oil, which is advisable for the synthesis of alkylated waste vegetable oil and aryl fatty acids derivatives.

Ethics. I/we confirm that we have read and comply with the Royal Society publishing ethics policy. This article does not present research with ethical considerations.

Data accessibility. The datasets supporting this article have been uploaded as part of the electronic supplementary material.

Authors' contributions. C.-L.G. and S.-Z.Y. designed the experiments, while C.-L.G. and X.W. conducted the experiments and wrote the paper. H.-Z.G. and J.-F.L. analysed the data. B.-Z.M. and S.-Z.Y. revised the paper. All the authors gave their final approval for publication.

Competing interests. We declare we have no competing interests.

Funding. This research was supported by the National Key Research and Development Program of China (grant no. 2017YFB0308900), the National Natural Science Foundation of China (grant no. 51574125), the Fundamental Research Funds for the Central Universities of China (grant no. 50321101917017) and the Research Program of State Key Laboratory of Bioreactor Engineering.

Acknowledgements. We are grateful for Teng-fei Shu and Zhou-qiang Yu, who provided the experimental materials during the research.

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
