## [Reviewer comments · Royal Society Open Science]

Review History

RSOS-192254.R0 (Original submission)

Review form: Reviewer 1

Is the manuscript scientifically sound in its present form?

No

Are the interpretations and conclusions justified by the results?

Yes

Is the language acceptable?

No

Do you have any ethical concerns with this paper?

No

Have you any concerns about statistical analyses in this paper?

No

Recommendation?

Reject

Comments to the Author(s)

Please see the attachment (Appendix A).

Review form: Reviewer 2

Is the manuscript scientifically sound in its present form?

Yes

Are the interpretations and conclusions justified by the results?

Yes

Is the language acceptable?

Yes

Do you have any ethical concerns with this paper?

No

Have you any concerns about statistical analyses in this paper?

No

Recommendation?

Accept with minor revision (please list in comments)

Comments to the Author(s)

This paper described the optimized condition producing alkylated vegetable oil through solid cation exchange resin and orthogonal analysis. The mechanism of catalyst dysfunction was also presented. Alkylated vegetable oil is very useful material which could be transferred into surfactants with highly interfacial activity. However similar zwitterionic surfactants were prepared through hydrolysis, alkylation, acyl chlorination, amidation and quaternization. The procedure, from waste vegetable oil to surfactants, should be shortened greatly through using alkylated vegetable oil as an intermediate. The mechanism of catalyst dysfunction was disclosed, which helpful for selection and pre-treatment of raw materials in production of bio-based surfactants. To sum up, I believe this paper is suitable for our readers and to be published in the journal.

There are some mistakes or confusing sentences in the paper. The text should be checked fully and corrected.

For instance, "the major factor causing the dysfunction the catalytic center" in Summary, how can we put "the dysfunction" and "the catalytic center" together? It might be "the dysfunction of the catalytic center" or "the catalytic center dysfunction"?

"through Diels-Alder cycloaddition reaction, triglycerides polymerize into polymers at high temperature" in Main text, "triglycerides polymerize into polymers"?

"Through addition of unused catalyst, the catalyst can maintain high catalytic efficiency" in Main text is confusing. Unused catalyst itself should be of highly catalytic efficiency.

Review form: Reviewer 3

Is the manuscript scientifically sound in its present form?

No

Are the interpretations and conclusions justified by the results?

No

Is the language acceptable?

No

Do you have any ethical concerns with this paper?

No

Have you any concerns about statistical analyses in this paper?

No

Recommendation?

Major revision is needed (please make suggestions in comments)

Comments to the Author(s)

This manuscript describes the use of a heterogeneous solid acid catalyst for the direct alkylation of waste vegetable oil. The reaction conditions have been optimized and the studies on deactivation mechanism was attempted. The paper is not well written and has some major problems to be addressed before it becomes publishable.

1. In the work, only conversion rates were reported and how about the product yields and selectivity? These data are necessary.
2. The authors tested three types of solid acids and deduced that pore capacity was a factor to consider when selecting the solid acid catalyst. There are many commercial solid acids in the market. To draw a feasible conclusion, it is suggested to test at least five solid acid catalysts with varied pore capacities.
3. In the study of deactivation mechanism, it is strongly recommended to characterize the solid acid catalysts before and after the reaction by using XPS, FTIR, elemental analysis, etc. to get information on structural changes.
4. It is recommended to re-examine the writing of the manuscript to make it more logical and clearer. "F significance", "As for waste vegetable oil like waste cooking oil", it was quite confusing. In addition, the quality of figures is low, please replace with high-res ones.
5. The heterogeneous catalyst deactivated after uses and it is thus not able to be well recycled. As a result, the significance of using heterogeneous catalysts has impaired. Discussions may be added on this part for future solutions, etc.

Decision letter (RSOS-192254.R0)

Dear Dr Gao:

Title: The optimization of heterogeneous catalytic conditions in the direct alkylation of waste vegetable oil

Manuscript ID: RSOS-192254

The editor assigned to your manuscript has now received comments from reviewers. We would like you to revise your paper in accordance with the referee and Subject Editor suggestions which can be found below (not including confidential reports to the Editor). Please note this decision does not guarantee eventual acceptance.

Please submit your revised paper before 05-Jun-2020. Please note that the revision deadline will expire at 00.00am on this date. If we do not hear from you within this time then it will be assumed that the paper has been withdrawn. In exceptional circumstances, extensions may be possible if agreed with the Editorial Office in advance. We do not allow multiple rounds of revision so we urge you to make every effort to fully address all of the comments at this stage. If deemed necessary by the Editors, your manuscript will be sent back to one or more of the original reviewers for assessment. If the original reviewers are not available we may invite new reviewers.

RSC Associate Editor:
Comments to the Author:
(There are no comments.)

RSC Subject Editor:
Comments to the Author:
(There are no comments.)

Reviewers' Comments to Author:

Reviewer: 1

Comments to the Author(s)

Please see the attachment

Reviewer: 2

Comments to the Author(s)

This paper described the optimized condition producing alkylated vegetable oil through solid cation exchange resin and orthogonal analysis. The mechanism of catalyst dysfunction was also presented. Alkylated vegetable oil is very useful material which could be transferred into surfactants with highly interfacial activity. However similar zwitterionic surfactants were prepared through hydrolysis, alkylation, acyl chlorination, amidation and quaternization. The procedure, from waste vegetable oil to surfactants, should be shortened greatly through using alkylated vegetable oil as an intermediate. The mechanism of catalyst dysfunction was disclosed, which helpful for selection and pre-treatment of raw materials in production of bio-based surfactants. To sum up, I believe this paper is suitable for our readers and to be published in the journal.

There are some mistakes or confusing sentences in the paper. The text should be checked fully and corrected.

For instance, "the major factor causing the dysfunction the catalytic center" in Summary, how can we put "the dysfunction" and "the catalytic center" together? It might be "the dysfunction of the catalytic center" or "the catalytic center dysfunction"?

"through Diels-Alder cycloaddition reaction, triglycerides polymerize into polymers at high temperature" in Main text, "triglycerides polymerize into polymers"?

"Through addition of unused catalyst, the catalyst can maintain high catalytic efficiency" in Main text is confusing. Unused catalyst itself should be of highly catalytic efficiency.

Reviewer: 3

Comments to the Author(s)

This manuscript describes the use of a heterogeneous solid acid catalyst for the direct alkylation of waste vegetable oil. The reaction conditions have been optimized and the studies on deactivation mechanism was attempted. The paper is not well written and has some major problems to be addressed before it becomes publishable.

1. In the work, only conversion rates were reported and how about the product yields and selectivity? These data are necessary.
2. The authors tested three types of solid acids and deduced that pore capacity was a factor to consider when selecting the solid acid catalyst. There are many commercial solid acids in the market. To draw a feasible conclusion, it is suggested to test at least five solid acid catalysts with varied pore capacities.
3. In the study of deactivation mechanism, it is strongly recommended to characterize the solid acid catalysts before and after the reaction by using XPS, FTIR, elemental analysis, etc. to get information on structural changes.
4. It is recommended to re-examine the writing of the manuscript to make it more logical and clearer. "F significance", "As for waste vegetable oil like waste cooking oil", it was quite confusing. In addition, the quality of figures is low, please replace with high-res ones.
5. The heterogeneous catalyst deactivated after uses and it is thus not able to be well recycled. As a result, the significance of using heterogeneous catalysts has impaired. Discussions may be added on this part for future solutions, etc.

Author's Response to Decision Letter for (RSOS-192254.R0)

See Appendix B.

RSOS-192254.R1 (Revision)

Review form: Reviewer 1

Is the manuscript scientifically sound in its present form?

Yes

Are the interpretations and conclusions justified by the results?

Yes

Is the language acceptable?

Yes

Do you have any ethical concerns with this paper?

No

Have you any concerns about statistical analyses in this paper?

No

Recommendation?

Accept as is

Comments to the Author(s)

NA

Review form: Reviewer 2

Is the manuscript scientifically sound in its present form?

Yes

Are the interpretations and conclusions justified by the results?

Yes

Is the language acceptable?

Yes

Do you have any ethical concerns with this paper?

No

Have you any concerns about statistical analyses in this paper?

No

Recommendation?

Accept as is

Comments to the Author(s)

The revised manuscript can be accepted after its revision.

Review form: Reviewer 3**Is the manuscript scientifically sound in its present form?**

Yes

Are the interpretations and conclusions justified by the results?

Yes

Is the language acceptable?

Yes

Do you have any ethical concerns with this paper?

No

Have you any concerns about statistical analyses in this paper?

No

Recommendation?

Accept as is

Comments to the Author(s)

Most of the problems have been properly addressed.

Decision letter (RSOS-192254.R1)

Dear Dr Gao:

Title: The optimization of heterogeneous catalytic conditions in the direct alkylation of waste vegetable oil

Manuscript ID: RSOS-192254.R1

It is a pleasure to accept your manuscript in its current form for publication in Royal Society Open Science. The chemistry content of Royal Society Open Science is published in collaboration with the Royal Society of Chemistry.

Yours sincerely,
Dr Laura Smith

Publishing Editor, Journals

RSC Associate Editor:
Comments to the Author:
(There are no comments.)

RSC Subject Editor:
Comments to the Author:
(There are no comments.)

Reviewer(s)' Comments to Author:
Reviewer: 2

Comments to the Author(s)
The revised manuscript can be accepted after its revision.

Reviewer: 3

Comments to the Author(s)
Most of the problems have been properly addressed.

Reviewer: 1

Comments to the Author(s)
NA

Appendix A

Journal: Royal Society Open Science

Manuscript ID: RSOS-192254

This manuscript by Gao and coworkers reports the alkylation of waste vegetable oil. The manuscript lacks scientific explanation for the obtained results. Please see following comments to improve the quality of the manuscript.

1. Please provide the details about the methods used to calculate conversions and yields shown in Table 1 and Table S1. What is the error and variability in the reported values? Experiments should be performed in triplicates and the errors in conversion and yield should be incorporated in Tables 1 and S1
2. Please explain in details why there is variations in the yields (Table S1) with change in catalyst loading, reaction temperature, and solvent ratio etc.
3. What is the method used to obtain the optimized reaction conditions shown in Table 1? What is the meaning of conversion rate? I missed the optimization data for the Amberlyst 35 and 36 catalysts. Please justify why different catalysts have different conversions. Please provide the detailed catalyst characterization: surface area, pore size, etc. Please correlate the catalyst characterization with the activity values.
4. Catalyst loading should be with respect to the volume of substrate (waste cooking oil) used in the reaction. It should not be with respect to the total reaction volume.
5. Fig. 2 should be moved to ESI, while Fig. S1 should be included in the main manuscript.
6. Fig. 1 is not legible, please redraw the structures using ChemDraw or another suitable tool.
7. The catalyst reusability and deactivation should be thoroughly explained (Fig S1 and Table S3).

As a conclusive final comment, manuscript has many grammatical errors, manuscript should be carefully checked for scientific discrepancies and should be rewritten much more concisely.

Appendix B

For Reviewer 1

This manuscript by Gao and coworkers reports the alkylation of waste vegetable oil. The manuscript lacks scientific explanation for the obtained results. Please see following comments to improve the quality of the manuscript.

1. Please provide the details about the methods used to calculate conversions and yields shown in Table 1 and Table S1. What is the error and variability in the reported values? Experiments should be performed in triplicates and the errors in conversion and yield should be incorporated in Tables 1 and S1

Response: The conversions were calculated by the change ratio of unsaturated fatty acid methyl esters before and after reactions using methyl stearate as the internal standard substance. The yields were calculated by the peak area (Extracted Molecular Ion Chromatogram) ratio of benzyl methyl stearate to the sum of benzyl methyl stearate and unsaturated fatty acid methyl esters. The calculation formulas of conversions and yields were added in Section 3.1 paragraph 2 (Revised ESM).

The error and variability of conversions and yields were added in the Table 1 (Revised Manuscript) and S1(Revised ESM).

$$\text{Conversion rate} = 1 - \frac{A'_{\text{unsaturated fatty acid methyl esters}}/A'_{\text{methyl stearate}}}{A_{\text{unsaturated fatty acid methyl esters}}/A_{\text{methyl stearate}}}$$

$$\text{Yield} = \frac{A''_{\text{benzyl methyl stearate}}}{A''_{\text{unsaturated fatty acid methyl esters}} + A''_{\text{benzyl methyl stearate}}}$$

Notes:

$A'_{\text{unsaturated fatty acid methyl esters}}$: The peak area of unsaturated fatty acid methyl esters in the GC-MS total ions chromatogram after reactions.

$A'_{\text{methyl stearate}}$: The peak area of methyl stearate in the GC-MS total ions chromatogram of after reactions.

$A_{\text{unsaturated fatty acid methyl esters}}$: The peak area of unsaturated fatty acid methyl esters in the GC-MS total ions chromatogram of waste vegetable oil.

$A_{\text{methyl stearate}}$: The peak area of methyl stearate in the GC-MS total ions chromatogram of waste vegetable oil.

$A''_{\text{unsaturated fatty acid methyl esters}}$: The peak area of unsaturated acid methyl esters in the GC-MS extracted molecular ion chromatogram after reactions.

$A''_{\text{benzyl methyl stearate}}$: The peak area of benzyl methyl stearate in the GC-MS extracted molecular ion chromatogram of after reactions.

2. Please explain in details why there is variations in the yields (Table S1) with change in catalyst loading, reaction temperature, and solvent ratio etc.

Response: According to the reviewer's recommendation, the variations in the yields (Table S1) with change in catalyst loading, reaction temperature, and solvent ratio were discussed in details in revised ESM.

"As showed in Table S1, the temperature has the greatest effect on the yield. The reaction time and catalyst loading also contributes a lot to the yield while the solvent ratio has the lowest effect on the yield. According to the Arrhenius equation, the reaction rate constant will increase with the rise of temperature, resulting in acceleration of reaction rate and variations in the yields. In addition, the activation energy will be significantly reduced after the introduction of catalyst, and the reaction rate was increased with the increase of catalyst loading. As for the solvent ratio, the introduction of solvent will influence the concentration of the raw materials, resulting in the variations of reaction rate and yields. As for the reaction time, the yields will be increased with the increase of reaction time until the reaction equilibrium is reached." has been inserted in Section 3.1(Revised ESM).

3. What is the method used to obtain the optimized reaction conditions shown in Table 1? What is the meaning of conversion rate? I missed the optimization data for the Amberlyst 35 and 36 catalysts. Please justify why different catalysts have different conversions. Please provide the detailed catalyst characterization: surface area, pore size, etc. Please corelate the catalyst characterization with the activity values.

Response: The optimized reaction conditions were obtained by comparing the conversion rate of five solid catalysts at their maximum operating temperature for 2 h. The conversion rate was calculated by the change ratio of unsaturated fatty acid methyl esters before and after reactions using methyl stearate as the internal standard substance. Moreover, the calculation formula of conversion rate was added in Section 3.1 paragraph 2 (Revised ESM). The optimization data of Amberlyst 35 and 36 catalysts were added in Table 1 (Revised Manuscript).

Capacity, maximum operating temperature and pore capacity of the five solid catalysts were provided in Table 1 (Revised Manuscript). The surface area of five solid catalysts was provided in Section 3.6.1(Revised ESM). Due to the structural differences of catalyst, different catalysts have different conversions and a conclusive paragraph" While the total exchange capacity of used catalysts was close to each other, the relationship between pore capacity and conversion rate was positive correlation. When

pore capacity was larger, the catalyst active center was able to contact much more substrates, the exchange of substances with reaction solutions was easy, and the reaction is performed fluently. " was inserted in Section 2 paragraph 5 (Revised Manuscript)

4. Catalyst loading should be with respect to the volume of substrate (waste cooking oil) used in the reaction. It should not be with respect to the total reaction volume.

Response: According to the reviewer's recommendation, the catalyst loading has been updated with respect to the volume of substrate (waste cooking oil) used in the reaction (Revised ESM) and the associated data (catalyst loading) in Table S1 Column "Catalyst loading" has been updated at the same time (Revised ESM)

5. Fig. 2 should be moved to ESI, while Fig. S1 should be included in the main manuscript.

Response: According to the reviewer's recommendation, Fig. 2(previous version) has been moved to revised ESM as Fig.S1, while Fig. S1(previous version) has been included in the revised manuscript as Fig. 2. In addition, associated figure captions have been updated in revised manuscript and revised ESM.

6. Fig. 1 is not legible, please redraw the structures using ChemDraw or another suitable tool.

Response: According to the reviewer's recommendation, Fig. 1 has been redrawn by using ChemDraw and the high resolution Fig.1 has been updated in revised manuscript.

7. The catalyst reusability and deactivation should be thoroughly explained (Fig S1 and Table S3).

Response: According to the reviewer's recommendation, the catalyst reusability and deactivation (Fig S1 and Table S3, previous version) was thoroughly explained in revised manuscript and revised ESM.

As for Fig.S1 (previous version), a paragraph "The catalyst in the alkylation was recycled for 20 rounds in the optimized condition to investigate the operational life of the catalyst. The catalytic efficiency decreased rapidly from 97.8% to 53.9% with the recycle times increasing (Fig.2). In addition, the linear fitting curve of the experiment data was $y=1.014-0.012x$ and the calculated half-life was 42.2 h, which is much

shorter than the half-life of the traditional strong acid cation exchange resins. Thus, the catalyst must be deactivated and experiments should be performed to explore the deactivation mechanism in the direct alkylation of the waste vegetable oil." was inserted in Section 2(Revised Manuscript) to explain the reusability of catalyst.

As for Table S3, a paragraph " The contents of metallic elements in used catalyst has grown much higher than before, indicating that corrosion of reactor (made of stainless steel 316) has occurred in the alkylation process. However, the contribution of the corrosion to the declination of the exchange capacity was only 0.127 mmol/g[H⁺] after calculation, about 2.56% of the total exchange capacity. Thus, the corrosion of reactor which caused about 2.56% loss of total exchange capacity was not the key factor causing deactivation of catalyst." was inserted in Section 3.3.1 (Revised ESM) to explain the deactivation of catalyst.

As a conclusive final comment, manuscript has many grammatical errors, manuscript should be carefully checked for scientific discrepancies and should be rewritten much more concisely.

Response: According to the reviewer's recommendation, the manuscript has been checked and rewritten in the revised manuscript. For example,

"element composition" was written as "elemental composition" in the revised manuscript.

"operation life" was written as "operational life" in the revised manuscript.

" be converted to " was written as "be converted into" in the revised manuscript.

"the major factor causing the dysfunction the catalytic center" was rewritten as "the major factor causing dysfunction of the catalytic center" in the revised manuscript.

"was proved to the major factor" was written as " was proved to be the major factor " in the revised manuscript.

"alkylated oil was an important intermediate " was written as " alkylated oil is an important intermediate " in the revised manuscript.

"The surface/interfacial properties were much better " was written as " The surface/interfacial properties are much better " in the revised manuscript.

" involved the pretreatment of waste vegetable oil " was written as " involves the pretreatment of waste vegetable oil " in the revised manuscript.

"catalyst in the tradition synthesis route was homogeneous catalyst" was written as " catalyst used in the traditional synthesis route is homogeneous catalyst " in the revised manuscript.

"The catalyst in the alkylation were recycled " was written as " The catalyst in the alkylation was recycled " in the revised manuscript

"which were much shorter than the half-life " was written as "which is much shorter than the half-life " in the revised manuscript

"Previous study [30-32] showed that " was written as "Previous studies [30-32] showed that" in the revised manuscript

" with an increasing of aromatic hydrocarbons " was written as " with an increase of aromatic hydrocarbons " in the revised manuscript

" for avoiding the formation of insoluble polymers " was written as " to avoid the formation of insoluble polymers " in the revised manuscript

"the hydrogenated oil will be an ideal raw material" was written as "the hydrogenated castor oil will be an ideal raw material" in the revised manuscript

" using strong acid exchange resins as catalyst in the optimized condition " was written as " using strong acid exchange resins as catalyst under optimized conditions " in the revised manuscript

"conducted the experiments and writhening the paper " was written as " conducted the experiments and written the paper " in the revised manuscript

For Reviewer 2

This paper described the optimized condition producing alkylated vegetable oil through solid cation exchange resin and orthogonal analysis. The mechanism of catalyst dysfunction was also presented. Alkylated vegetable oil is very useful material which could be transferred into surfactants with highly interfacial activity. However similar zwitterionic surfactants were prepared through hydrolysis, alkylation, acyl chlorination, amidation and quaternization. The procedure, from waste vegetable oil to surfactants, should be shortened greatly through using alkylated vegetable oil as an intermediate. The mechanism of catalyst dysfunction was disclosed, which helpful for selection and pre-treatment of raw materials in production of bio-based surfactants. To sum up, I believe this paper is suitable for our readers and to be published in the journal.

There are some mistakes or confusing sentences in the paper. The text should be checked fully and corrected.

For instance, "the major factor causing the dysfunction the catalytic center" in Summary, how can we put "the dysfunction" and "the catalytic center" together? It might be "the dysfunction of the catalytic center" or "the catalytic center dysfunction"?

"through Diels-Alder cycloaddition reaction, triglycerides polymerize into polymers at high temperature" in Main text, "triglycerides polymerize into polymers"?

"Through addition of unused catalyst, the catalyst can maintain high catalytic efficiency" in Main text is confusing. Unused catalyst itself should be of highly catalytic efficiency.

Response: Thank you for reminding of inaccurate expressions or illustrations in the manuscript and the mistakes has been corrected in the revised manuscript.

"the major factor causing the dysfunction the catalytic center" was rewritten as "the major factor causing dysfunction of the catalytic center" in the revised manuscript.

"through Diels-Alder cycloaddition reaction, triglycerides polymerize into polymers at high temperature" was rewritten as "through Diels-Alder cycloaddition reaction between triglycerides, polymers were formed at high temperature" in the revised manuscript.

"Through addition of unused catalyst, the catalyst can maintain high catalytic efficiency" was deleted in the revised manuscript.

For Reviewer 3

This manuscript describes the use of a heterogeneous solid acid catalyst for the direct alkylation of waste vegetable oil. The reaction conditions have been optimized and the studies on deactivation mechanism was attempted. The paper is not well written and has some major problems to be addressed before it becomes publishable.

1. In the work, only conversion rates were reported and how about the product yields and selectivity? These data are necessary.

Response: According to the reviewer's recommendation, the associated data of product yields and selectivity were added in the Table 1 Column "Yield" and "Selectivity" (Revised Manuscript). In addition, the calculation formula of the selectivity and yield was added in Section 3.1(Revised ESM).

2. The authors tested three types of solid acids and deduced that pore capacity was a factor to consider when selecting the solid acid catalyst. There are many commercial solid acids in the market. To draw a feasible conclusion, it is suggested to test at least five solid acid catalysts with varied pore capacities.

Response: According to the reviewer's suggestion, in the catalyst selection part, the results of two newly investigated solid acid catalysts (HND 2 and Amberlyst 15)

were added to Table 1 in the revised manuscript. It is helpful to draw a feasible conclusion that five solid acid catalysts with varied pore capacities were compared

3. In the study of deactivation mechanism, it is strongly recommended to characterize the solid acid catalysts before and after the reaction by using XPS, FTIR, elemental analysis, etc. to get information on structural changes.

Response: According to the reviewer's suggestion, the structural changes of the solid acid catalysts before and after the reaction was characterized using XPS, FTIR, elemental analysis. Associated data were listed and discussed in revised manuscript and revised ESM.

As for FTIR, a paragraph "As for -CH₂-, the transmittance had been greatly increased for the catalyst after uses. In addition, the absorption of C=O indicated that new substances were attached to the catalyst after uses. Above all, some fatty acid derivatives might be attached to the catalyst" was inserted in Section 3.6.3 (Revised ESM). Moreover, a sentence "As a consequence, some extra substances attached to the catalyst were the major factor causing the dysfunction of the catalytic center, which was also confirmed by FTIR results (ESM, Fig.S5, Fig.S6)" was inserted in Section 2 paragraph 9 (Revised Manuscript).

As for XPS, a paragraph "It could be inferred from the XPS results that there were C/O/S/Fe/Cr/Ni/Mo/Mn elements in the solid catalyst. Moreover, the peak of sulfur element has not changed significantly, indicating that the catalytic center has not lost greatly and the loss of catalytic center may not be the major factor causing the catalyst deactivation" was inserted in Section 3.6.4 (Revised ESM)

As for elemental analysis, a paragraph "In addition, the content of sulfur had decreased significantly from 16.7% to 11.9% (ESM, table S5). Thus, some extra substances were attached to the catalyst causing the declination of exchange capacity and the drop of the sulfur content" was inserted in Section 2 paragraph 9 (Revised Manuscript)

4. It is recommended to re-examine the writing of the manuscript to make it more logical and clearer. "F significance", "As for waste vegetable oil like waste cooking oil", it was quite confusing. In addition, the quality of figures is low, please replace with high-res ones.

Response: According to the reviewer's recommendation, the manuscript has been re-examined and changed. The confusing expressions were rewritten and the figures

(including Fig.1, Fig.2 and Fig.3) were replaced with high-resolution ones in the revised manuscript.

"F significance tests" was rewritten as "F-test (variance ratio)" in revised manuscript;

"As for waste vegetable oil like waste cooking oil" was rewritten as "As for waste cooking oil" in the revised manuscript.

"the major factor causing the dysfunction the catalytic center" was rewritten as "the major factor causing dysfunction of the catalytic center" in the revised manuscript.

"through Diels-Alder cycloaddition reaction, triglycerides polymerize into polymers at high temperature" was rewritten as "through Diels-Alder cycloaddition reaction between triglycerides, polymers were formed at high temperature" in the revised manuscript.

"Through addition of unused catalyst, the catalyst can maintain high catalytic efficiency" was deleted in the revised manuscript.

"C, H, metallic elements contents of the catalyst" was written as "metallic and non-metallic (C, H) elements contents of the catalyst" in the Section 1 (Revised Manuscript)

"ESI, GC-MS, PyGC-MS" was written as "electrospray ionization mass spectrometry, gas chromatography mass spectrometry, pyrolysis-gas chromatography/mass spectrometry" in the Section 1 (Revised Manuscript)

"ICP" was written as "ICP (Inductively Coupled Plasma Atomic Emission Spectrometer)" in the revised manuscript.

"low in the contents of polyunsaturated fatty acids" was written as "having low concentration of polyunsaturated fatty acids" in the Section 1 (Revised Manuscript)

"produced on large scale" was deleted in the revised manuscript.

"As a consequence" was written as "Consequently" in the revised manuscript.

"the optimized alkylation condition was achieved" was written as "the following optimized alkylation conditions were achieved" in the revised manuscript.

5. The heterogeneous catalyst deactivated after uses and it is thus not able to be well recycled. As a result, the significance of using heterogeneous catalysts has impaired. Discussions may be added on this part for future solutions, etc.

Response: The heterogeneous catalyst deactivation mechanism was revealed in the manuscript. In addition, solutions were put forward to avoid or reduce deactivation of the heterogeneous catalyst. Hydrogenated castor oil and waste vegetable oil with

high abundance of oleic acid was recommended as raw material in the direct alkylation of waste vegetable oil. According to the reviewer's recommendation,

Four paragraphs " Diels-Alder cycloaddition reaction between triglycerides was based on the polyunsaturated fatty acids in the waste vegetable oil. In order to achieve high conversions and maintain excellent catalytic efficiency, waste vegetable oil was divided into three categories according to the contents of polyunsaturated fatty acids.

As for waste cooking oil, the contents of polyunsaturated fatty acids were very high, resulting in formation of insoluble polymers and declination of the catalytic efficiency as well as operational life. Two-step alkylation is necessary for this kind of waste vegetable oil to avoid the formation of insoluble polymers. Through hydrolysis or transesterification, waste vegetable oil can be converted to fatty acid derivatives which polymerize into soluble dimers and do not influence the operational life of the catalyst. For example, the half-life of the catalyst can be extended from 42.2 h to 453 h (ESM, Fig. S4) when using transesterification products of the waste vegetable oil in the preparation of alkylated waste vegetable oil.

As for olive oil, the content of oleic acid was high while the contents of polyunsaturated fatty acids were relatively low, resulting in mild formation of insoluble polymers and slow declination of the catalytic efficiency as well as operational life.

As for hydrogenated castor oil, 12-hydroxystearic acid could be used in the alkylation of hydrogenated castor oil, avoiding the formation of polymers. Thus, the hydrogenated castor oil will be an ideal raw material in the direct alkylation of waste vegetable oil." were provided in Section 2 (Revised Manuscript)